# Hardware-In-The-Loop Simulations of Hole/Crack Identification in a Composite Plate

**DOI:** 10.3390/ma13020424

**Published:** 2020-01-16

**Authors:** Yen-Chu Liang, Yun-Ping Sun

**Affiliations:** 1Department of Aeronautics and Astronautics, R.O.C. Air Force Academy, Gangshan 820, Taiwan; 2Department of Mechanical Engineering, Cheng Shiu University, Kaohsiung 830, Taiwan; ypsun@gcloud.csu.edu.tw

**Keywords:** hardware-in-the-loop simulations (HILS), hole/crack identification, multiple loading modes, nonlinear optimization, structural health monitoring (SHM)

## Abstract

The technology of hardware-in-the-loop simulations (HILS) plays an important role in the design of complex systems, for example, the structural health monitoring (SHM) of aircrafts. Due to the high performance of personal computers, HILS can provide practical solutions to many problems in engineering and sciences, especially in the huge systems, giant dams for civil engineering, and aircraft system. This study addresses the HILS in hole/crack identification in composite laminates. The multiple loading modes method is used for hole/crack identification. The signals of strains measured from the data-acquisition (DAQ) devices are accomplished by the graphical software LabVIEW. The results represent the actual responses of multiple loading mode tests of real specimens. A personal computer is employed to execute the identification work according to the strain data from DAQ devices by using a nonlinear optimization approach. When all the criteria are satisfied, the final identification results will be obtained. HILS will achieve real time identification of hole/crack in the composite plate by using the actual response measured from the sensors. Not only the size, but also the location and orientation of the crack/hole in a composite plate are successfully identified herein.

## 1. Introduction

Studies on structural health monitoring (SHM) are popular because the significant development of computer power, data processing capability, and modern sensor technology make real time detection of fractures possible. The various types of sensors are installed onto or embedded in the object’s structure and mass data related to the object’s certain conditions are automatically collected [1,2,3]. The safety of the object is determined by the analyses of the collected information. Aircraft structural maintenance is one of the critical operations to guarantee continued airworthiness. Reducing the cost and increasing the safety of air transportation are the main goals in the massive aviation business. The characteristics of high strength and low density for composite materials have highly increased the percentage of composites in aircraft structures. However, composite structures, compared to metallic structures, have more complex damage modes because of their anisotropic properties. X. Chen et al. performed a statistical analysis of Chinese airline maintenance departments in 2012. The records of wing structural damages for two types of Boeing aircraft fleet in a 10-year period were obtained. Dents are the most frequent damage mode (38%), followed by paint peeling off (24%). The damage mode for wind erosion is 12%, 2% for hole damage, and 8% for delamination [3]. Therefore, the detection of defects, such as cracks, holes, and delamination in composite structures is increasingly important during maintenance. In order to enhance the safety of structures and reduce the casualties or property losses, the correct and efficient methods to build the monitoring system of the aircraft health are worthy of research topics. 

There are two main fields in damage detection. One field is applying the dynamic wave to the structures to compare the differences between healthy and nonhealthy structures [4,5,6,7,8,9,10,11,12,13]. The other field is the identification of defects by using static response [14,15,16,17,18,19,20,21,22], that is applying directional loading to the structures. In the dynamic methods, Kessler from MIT started damage detection in composite materials by using Lamb wave methods, and many researchers followed the studies and achieved advanced results. The computational time, number of sensors, and size of the detected object increased from 12 to 16 h, from 4 to 240 sensors, and 60 × 60 to 225 × 300 cm^2^ for size, respectively. Chen et al. achieved precise crack identification in cantilever beam, up to 2% in location detection and 4% in size detection by using the first three natural frequencies [10]. In 2015, Matarazzo et al. proposed that today’s SHM procedures were suitable for tomorrow’s big data [12]. The trends of online SHM proliferate due to the development of future technology [13]. On the other hand, an inexpensive method of defect detection for composite materials in real time will play an important role in the maintenance, repair, and overhaul of aircrafts. Studies concerning the static methods have the advantage of cheap sensors and undisturbed signals compared to frequency. Crack or hole detection of beam or plate by using static measurements, strains, or displacements, can be found in [14,15,16,17,18,19,20,21,22]. Not only the finite element method (FEM), but also the boundary element method (BEM) or any forward analysis method is employed in these papers. Qiao et al. caught the location and size of delamination in composite laminate by using the experiment and finite element method to derive the dynamic and static measured information [20]. The problem of crack’s nonsensitivity to the global strain field can be solved by embedding fiber optic sensors into the composite laminates [22,23]. Hattori and Sáez [21] combined the methods of neural networks, self-organizing algorithms, and boundary element method to identify the cracks in magnetoelectroelastic materials. Okabe and Yashiro performed the hole detection by using periodic static loading [22]. These methods are similar to the studies we proposed in [15,16]. Thus, it is worthwhile to carry out advanced experiments on hole/crack identification by using static information.

The advantages of hardware-in-the-loop simulation (HILS) are short development time, lower cost of real experiments, and decreased danger during experiments, especially for the applications with high safety considerations. The engineers can efficiently revise the design of portal type systems by using the results of HILS. HILS is a perfect test method for the study of huge experimental data, for example the control systems for aircrafts [24,25]. In this study, a hole/crack in a square plate with layered laminate carbon composites is identified by using the strains measured from the inner boundary sensors. The HILS adopt a material test stand (MTS) to perform the multiple loading modes, a series of convenient and inexpensive strain gauges connected to the DAQ cards (NI9237) obtain the signals of sensors from four personal computers, and a nonlinear optimal identification program is executed in a personal computer. The strains measured from eight sensors under different loading modes are forwarded into the nonlinear optimization program to satisfy the criterion of convergence. The breakthrough point of the study is the change of loading mode forcing the optimal result of each step to serve as an initial guess for the next step under a different loading mode until the final result satisfies the convergence criterion. The contribution of the work is building the HILS system to realize the hole/crack identification of a composite plate by using multiple loading modes. The measured strains of plate subjected to real loading are the sources of object function of nonlinear optimization. The novelty of HILS is combining the multiple loading modes and the strains from the static loadings to improve the convergence in nonlinear optimization.

## 2. Anisotropic Composite Plates

There were many successful simulation results of identifications of hole or crack in a composite plate by using Stroh’s formula and the boundary element method [15]. The point load with infinity solutions of hole or crack in a composite laminate has been performed by Stroh’s formula [26]. The finite domain solutions of displacements, strains, stresses, or stress intensity factors are given by using the boundary element method. This field is the research of forward analysis. The stresses or strains sensitivities with respect to hole or crack sizes, locations, or orientations have been shown in [15]. Large variation of sensitive objective function is the key solution to find the size, location, and orientation identification of a crack or hole in the inverse problem. A general solution satisfying the basic equations of strain-displacement, stress-strain, and equilibrium for a two-dimensional anisotropic linear elastic medium has been presented as:(1)$$u={\left[\begin{array}{ccc} u_{1} & u_{2} & u_{3} \end{array}\right]}^{T}=Af\left(z\right)+\bar{A}\bar{f}\left(z\right)$$
(2)$$\phi ={\left[\begin{array}{ccc} \phi_{1} & \phi_{2} & \phi_{3} \end{array}\right]}^{T}=Bf\left(z\right)+\bar{B}\bar{f}\left(z\right)$$
where
(3)$$A=\left[\begin{array}{ccc} a_{1} & a_{2} & a_{3} \end{array}\right]$$
(4)$$B=\left[\begin{array}{ccc} b_{1} & b_{2} & b_{3} \end{array}\right]$$
(5)$$f\left(z\right)={\left[\begin{array}{ccc} f_{1}\left(z_{1}\right) & f_{2}\left(z_{2}\right) & f_{3}\left(z_{3}\right) \end{array}\right]}^{T}$$
(6)$$z_{\alpha }=x_{1}+p_{\alpha }x_{2},\;\alpha =1,2,3$$
u and ϕ are 3 × 1 column vectors denoting the displacements *u_1_*, *u_2_*, *u_3_* and stress functions $\phi_{1}$, $\phi_{2}$, $\phi_{3}$. Strains can be obtained from $\varepsilon_{ij}=\frac{1}{2}\left(u_{i,j}+u_{j,i}\right)$. ***A*** and ***B*** are eigenvectors of the material properties. The material eigenvalues $p_{\alpha }$ and eigenvectors $a_{\alpha }$, $b_{\alpha }$ are determined by the following eigenrelations:(7)Nξ=pξ,
(8)$$N=\left[\begin{array}{cc} N_{1} & N_{2}\\ N_{3} & N_{1}^{T} \end{array}\right]$$
(9)$$\xi =\left[\begin{array}{c} a\\ b \end{array}\right],$$
(10)$$N_{1}=-T^{-1}R^{T},$$
(11)$$N_{2}=T^{-1}=N_{2}^{T},$$
(12)$$N_{3}=RT^{-1}R^{T}-Q=N_{3}^{T},$$
and
(13)$$Q_{ik}=C_{i1k1},$$
(14)$$R_{ik}=C_{i1k2},$$
(15)$$T_{ik}=C_{i2k2}.$$
$C_{ijks}$ are the elastic constants which are assumed to be fully symmetric and positive definite. Detailed descriptions can be found in [26]. The Green’s function of an infinite anisotropic plate containing a traction-free hole subjected to a point force $\hat{P}$ applied at point $\hat{x}$ is expressed as:
(16)$$f\left(z\right)=\frac{1}{2\pi i}\langle \log(\varsigma_{\alpha }-{\hat{\varsigma }}_{\alpha })\rangle A^{T}\hat{P}+\overset{3}{\underset{k=1}{\sum }}\frac{1}{2\pi i}\langle \log(\varsigma_{\alpha }^{-1}-{\bar{\hat{\varsigma }}}_{k})\rangle B^{-1}\bar{B}I_{k}{\bar{B}}^{T}\bar{\hat{p}}$$
(17)$$\varsigma_{\alpha }=\frac{z_{\alpha }+\sqrt{z_{\alpha }^{2}-a^{2}-p_{\alpha }^{2}b^{2}}}{a-ip_{\alpha }b}$$
(18)$${\hat{\varsigma }}_{\alpha }=\frac{{\hat{z}}_{\alpha }+\sqrt{{\hat{z}}_{\alpha }^{2}-a^{2}-p_{\alpha }^{2}b^{2}}}{a-ip_{\alpha }b}$$
*a*, *b* are the lengths of the semi-axes of the ellipse. For a straight crack, let *b* = 0. Equation (5) is the special fundamental solution of boundary element formulation. Consider the real structural problem; the finite domain solution can be obtained by using the boundary element method. This part has been completed by Hwu and Liang [15]. For the crack problem, it is always interesting to know the stress intensity factors at the crack tip which are defined as [27]
(19)$$K=\left\{\begin{array}{c} K_{II}\\ K_{I}\\ K_{III} \end{array}\right\}=\underset{r\to 0}{\lim}\sqrt{2\pi r}\sigma_{2}$$
where *r* is the distance ahead of the crack tip and can be obtained from stress functions shown in Equation (2). The stress intensity factors can be expressed in terms of remote boundary displacements and tractions [28]. Figure 1 shows the profile of a hole in a square composite plate. The conditions of loading and fixed support and the sensor locations are shown in the figure. The position of hole center is (x, y). Hole size are 2*a* and 2*b*. *θ* is the hole orientation. There are three loading modes in the problem: Open, shear, and tear modes, as shown in Figure 2. In practice, the hole or crack located in the plate will not be always horizontal. We cannot predict the orientation of the hole/crack. We cannot produce pure loading modes. If the pure loading mode is not easily actuated, any kind of independent mixed loading modes can be used. It is more reasonable to give the applied loadings in the directions of X, Y, or Z in Figure 1. Therefore, we use the concept of open, shear, and tear modes and apply the mixed loading modes to identify the hole/crack.

## 3. Nonlinear Optimization

There are some sensitivity analyses before starting the identification of hole or crack parameters. The variations of normal strain and shear strain functions with respect to the hole/crack size location and orientation are shown in [15]. The aim of this paper is to demonstrate an experiment on hole/crack identification in a composite plate by using measured information from static loading. The strains measured from the nonlinear optimization of the problem can be defined as:(20)$$\alpha_{i}^{kM}=\alpha_{i}{\left(a,\;y,\;\theta ,\;a,\;b,\;s_{k}\right)}_{M}$$
where *S_K_* denotes the sensor position shown in Figure 1 (k = 1–8). The lower index *_M_* is the type of loading mode (_M_ = 1–3, as shown in Figure 2). *i* is the selected strain when *i* = 1 means normal strain $\varepsilon_{xx}$; *i* = 2 means shear strain $\varepsilon_{xy}$ and *i* = 3 signifies normal strain $\varepsilon_{yy}\;$ for loading modes *_M_* = 1 or 2. *i* = 1 denotes shear strain $\varepsilon_{xz}$ and *i* = 2 denotes shear strain $\varepsilon_{yz}$ when the loading mode *_M_* = 3. The *x*, *y*, *a*, *b*, and *θ* are the parameters of hole or crack which need to be identified. The insensitivities of the static strains to the hole/crack geometry and location may be overcome by the multiple loading modes. We may switch the loading condition to another loading mode and improve the search for the hole/crack geometry and location when the hole/crack cannot be identified by the static strains under a certain single loading condition. Repeat the process until the convergence criterion is satisfied. The multistep nonlinear optimization can be designed as follows.

Minimize objective function, $\Phi_{M}$:(21)$$\Phi_{M}=\overset{L}{\underset{k=1}{\sum }}\sum_{i=1}^{3}\left(\frac{\alpha_{i}^{kM}}{{\bar{\alpha }}_{i}^{kM}}-1\right){}_{k}^{2}$$
For loading mode *_M_* (= 1–3) subject to: (22)$$g_{i}=\sum_{k=1}^{L}\left(\frac{\alpha_{i}^{kM}}{{\bar{\alpha }}_{i}^{kM}}-1\right){}_{k}^{2}-e<0,\;i=1,2,3.$$
${\bar{\alpha }}_{i}^{kM}$ is the strain value of the reference problem. To avoid any numerical ill conditions, the value of $\frac{\alpha_{i}^{kM}}{{\bar{\alpha }}_{i}^{kM}}$ should be normalized to be on the order of unity. *L* is the number of sensors (*L* = 8 in Figure 1). *e* is the error tolerance of convergence which can be decreased in each step, as shown in the flowchart of Figure 3. The upper bounds and the lower bounds of designed variables are:(23)$$a_{l}<a<a_{u},$$
(24)$$b_{l}<b<b_{u},$$
(25)$$x_{l}<x<x_{u},$$
(26)$$y_{l}<y<y_{u},$$
(27)$$\theta_{l}<\theta <\theta_{u}.$$
(28)max(acos|θ|,bsin|θ|)−min(x, W−x)<0,
(29)max(asin|θ|,bcos|θ|)−min(y, H−y)<0
*W*, *H* are width and height of the square plate, respectively. The upper bounds of *a* and *b* are chosen to be not too large: $a_{u}$ = $b_{u}$ = *W*/4. The lower bounds of *a* and *b* are chosen to be a very small value 10^−6^ because the crack is the special case of elliptical hole when *b* approaches zero. To prevent a multi-value representation, $\theta_{u}=\pi /2$ and $\theta_{l}=-\pi /2$. The location bounds are defined as: $x_{l}=y_{l}=d$ and $x_{u}=W-d,\;\;y_{u}=H-d$ if the sensor positions locate a distance *d* from the side of the plate. Equations (28) and (29) express the conditions to enforce the hole or crack inside the plate. This important program is the identification of hole/crack size *a, b*, location *x, y,* and orientation *θ* which will be one system in the hardware-in-the-loop. Figure 3 shows the flowchart of nonlinear optimization with multiple loading modes.

The detailed simulations of hole/crack identification by static strains from multiple loading modes can be found in [15]. To save the cost and get the optimal design of HILS, many simulations should be done. According to the simulation results of [15], four, eight, twelve, and sixteen sensors were designed. The range of sensor spacing to the plate width are 0.8, 0.4, 0.267, and 0.2, and the maximum errors of small hole (2*a/W* = 0.044, *2b/W* = 0.022) identified results are 17.1%, 4.7%, 8.86%, and 4.5%, respectively. The proper ratio of sensor spacing to the plate width is 0.4. Hole/crack identifications of various materials, such as isotropic, anisotropic, piezoelectric materials were accomplished and the maximum error is 14.1% for piezoelectric material. Finally, the critical size of the hole/crack is suggested to be 2*a/W* = 0.03. The experiments of HILS are based on the provided information to manufacture the tested specimen and arrange the locations and number of sensors.

## 4. Hardware-In-The-Loop Simulations

In this paper, the carbon composite plates with an elliptical hole or a crack need to be made previously. The fiber orientations of six layers of carbon are [0/90/0]_s_. There are isotropic, orthotropic, anisotropic, and piezoelectric materials designed in our nonlinear optimization programs. Here, we choose the quasi-isotropic plate not only is it easy to manufacture but also it can reduce the sources of errors. The size of the plate is 30 × 30 cm^2^. The heat-compressor machine and a composite plate with an elliptical hole are shown in Figure 4. The structure of HILS in this paper is shown in Figure 5. There are three steps in the HILS.


*Step 1:*


The material test stand of multiple loading modes (loading mode I and loading mode II) is shown on the left and the middle of Figure 6. There are eight sensor positions in this plate. The type of strain gauge is KFG-5-120-C1-11 produced by KYOWA electronic instruments. The DAQ card NI9237 with four channels is shown on the right-hand side of Figure 6. The loading time profile is shown in Figure 7. The designed maximum load of the MTS is 5 kN and the stop loading in this paper is 1 kN. 


*Step 2:*


Usually, there are eight sensor positions, as shown in Figure 1. In Equation (20), strains $\varepsilon_{xx}$, $\varepsilon_{xy}$, and $\varepsilon_{yy}$ for loading mode I or II and strains $\varepsilon_{xz}$ and $\varepsilon_{yz}$ for loading mode III are required in each sensor. There will be 24 channels of strain measurement in loading modes I and II, and 16 channels of strain measurement in loading mode III. As it makes no difference whether the strain gauges are embedded into the plate, the strain gauges were set on the top surface of the plate. The sensors embedded in the plate for mode III are a problem. The plate is considered as plane stress for thin thickness. The fractures may generate during embedding the strain gauges into the plate. Therefore, only loading modes I and II is considered in this paper. To reduce the cost of future development, the strain $\varepsilon_{xy}$ is eliminated to leave only the 16 channels (four DAQ cards) required. Figure 8 and Figure 9 show the execution of strain measurement in the composite plate. 


*Step 3:*


The core process in the HILS is the program of nonlinear optimal identification in the personal PC, as shown in Figure 5. The algorithm has been described in Section 2 and Section 3. The real response of the composite plate with a hole/crack subjected to loading mode I or II is forwarded into the nonlinear optimization program. The program starts from an initial guess of hole/crack size, location, and orientation (*a, b, x, y,*
*θ*) under the situation of loading mode I or II and stops at the conditions when all the criteria of Equations (22)–(29) are satisfied. If the criteria are not satisfied, the optimal search will go back to another loading mode by using the final result of the previous loading mode to be the initial guess, and execute the loop till the program is convergent.

## 5. Results and Discussion

The test plate is a six-layered [0/90/0]_s_ carbon plate with 300 × 300 mm^2^ dimension. As shown in Figure 1, 8 sensors are located at the inner square boundary at 27 × 27 cm^2^. The material properties of the unidirectional carbon fiber prepreg are: Young’s modulus E_1_ = 160.8 GPa, E_2_ = 11.9 GPa, Poisson’s ratio ν = 0.326, shear modulus G = 6.8 GPa, and density ρ = 1530 kg/m^3^. 

### 5.1. Hole Identification

The parameters of the hole are *a =* 9 mm, *b =* 6.5 mm, *x =* 200 mm, *y =* 153 mm, and *θ =* 45°. The strains calculated from the numerical solutions of the boundary element method are shown in Table 1. Table 2 shows the strains measured in the HILS. In the bottom of Table 2, the value of $\Phi_{M}$ indicates the differences between the numerical data (Table 1) and experiment data (Table 2). It implicates that $\Phi_{M}$ is small. The values of *ε_xx_* and *ε_yy_* in Table 1 and Table 2 are close. The objective function $\Phi_{M}^{1}$ is calculated by a different hole. The large value of $\Phi_{M}^{1}$ means the objective function of the different hole is sensitive. The identified results of the hole by using the multiple loading modes are presented in Table 3. The errors increase when identifications do not use only pure simulations. The source of errors may be the loosened jigs of MTS. The jigs were sometimes slightly loosened when the plate was removed. The other error source may be the installations of the strain gauges. The identified procedures are plotted in Figure 10. IG means initial guess. TG means target. BEM means identifying a hole by using pure simulations. HILS is the identification results of a hole by using HILS. No. 1 is the identified results by setting the error tolerance *e* in Equation (6) 1 (mode I). The error tolerance *e* = 0.5 (mode II) for No.2 and *e* = 0.1 (mode I) for No.3. The sequence convergence criterion of error tolerance *e* in Equation (6) is 1 (mode I)→0.5 (mode II)→0.1 (mode I)→0.05 (mode II)→0.01 (mode I)→0.005 (mode II)→0.001 (mode I)→0.0005 (mode II). The final result of each loading mode is the initial guess of the next loading mode until the convergence criterion is satisfied. The number marked in Figure 10 signifies these seven identified processes. 

### 5.2. Crack Identification

It is well known that crack identification is more difficult than hole identification because of the nonsensitive response of crack parameters to the strains around the plate [15]. Table 4 and Table 5 show the strains calculated from BEM and the strains obtained from the experiment, respectively. The objective function of another crack is presented in the last line of Table 5. The identified results by using the numerical method or HILS are shown in Table 6 and plotted in Figure 11. IG means initial guess. TG means target. BEM means identifying a crack by using pure BEM. HILS is the identification results of a crack by using HILS. The crack identification by using HILS is not good. The errors increase to 22.9% for identified orientation, 21.48% for size identification, and 18.97% for vertical position identification; this may result from the open mode effect of a larger crack with larger orientation being the same as the one of a smaller crack with smaller orientation. In shear mode, a vertical crack is more sensitive than a horizontal crack. On the other hand, the design of jig support may be improved by some fixtures to completely clamp the plate. The sequence convergence criterion of error tolerance *e* is 1 (mode I)→1 (mode II)→0.5 (mode I)→0.5 (mode II)→0.1 (mode I)→0.05 (mode II)→0.01 (mode I)→0.005 (mode II)→0.001 (mode I)→0.0005 (mode II)→0.0005 (mode I).

## 6. Conclusions

Strains are the most convenient data to acquire in a static state. This paper demonstrates the feasibility of identifying a hole/crack by using only the strains measured from the composite plate. The experiment method is the hardware-in-the loop simulations (HILS). HILS include: A material test stand to give multiple loadings and fixed support to the composite plate having a hole/crack inside, four DAQ cards to acquire the strains around the inner plate boundary, a well-developed nonlinear optimization program to stably search the global minimum results. The experiment of hole/crack identification by using multiple loading modes is performed and the identified results of hole/crack size, location, and orientation are satisfied. The errors of HILS (from the experiments) are larger than the ones of BEM (from the simulations). Moreover, any kind of independent mixed loading mode can be applied if the pure loading mode is not easily actuated. This conforms to the real loading situations of structures. The accuracy may be increased when the independent mixed loading modes are added into HILS. 

## Figures and Tables

**Figure 1 materials-13-00424-f001:**
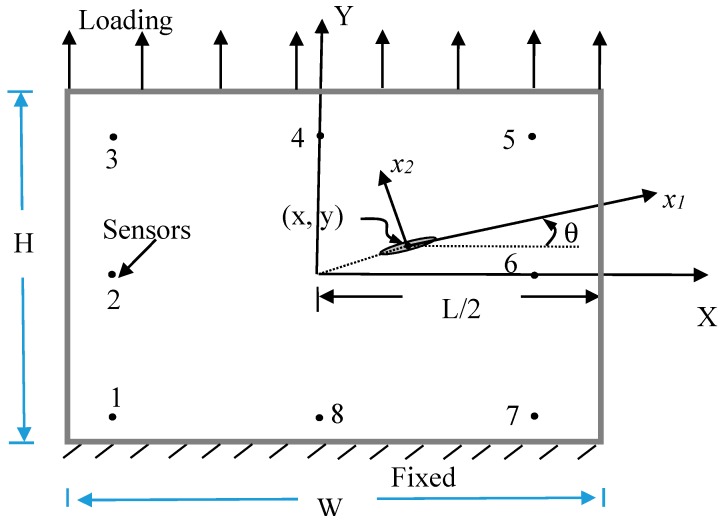
Loading and fixed conditions of a hole in the square composite plate and the sensor positions (no. 1~8).

**Figure 2 materials-13-00424-f002:**
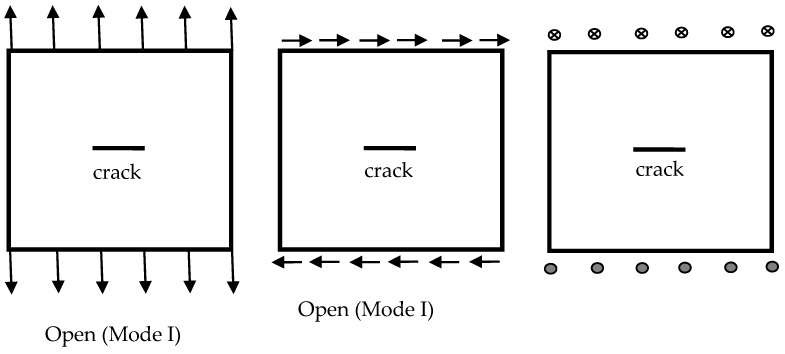
Loading modes of a crack in the square composite plate.

**Figure 3 materials-13-00424-f003:**
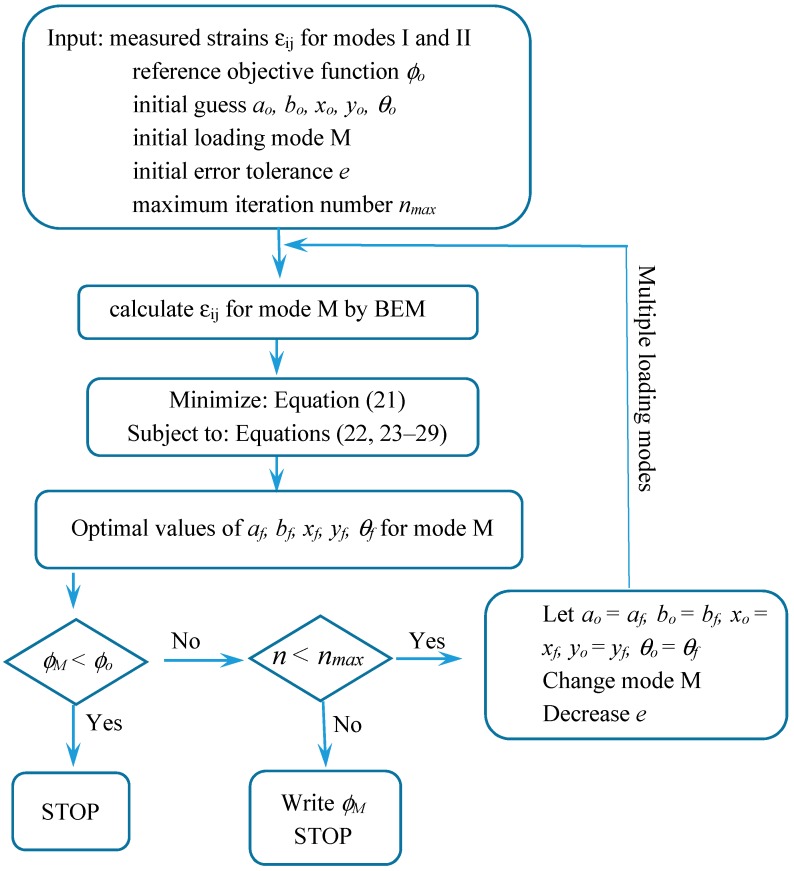
Flowchart of nonlinear optimization with multiple loading modes.

**Figure 4 materials-13-00424-f004:**
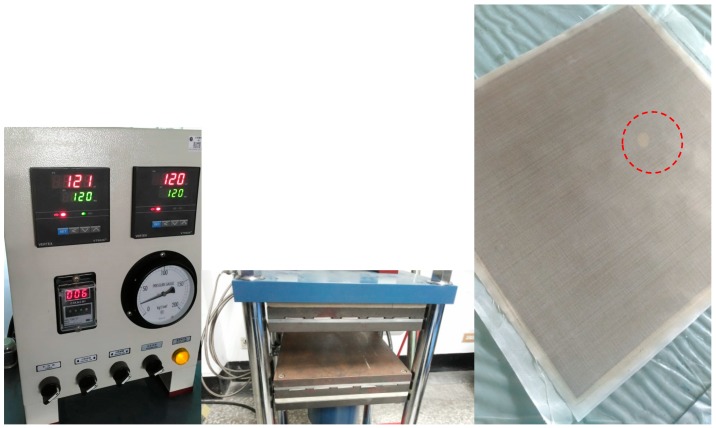
Equipment of composite plate manufacturing (**left**) and the plate with a hole marked in red circle (**right**).

**Figure 5 materials-13-00424-f005:**
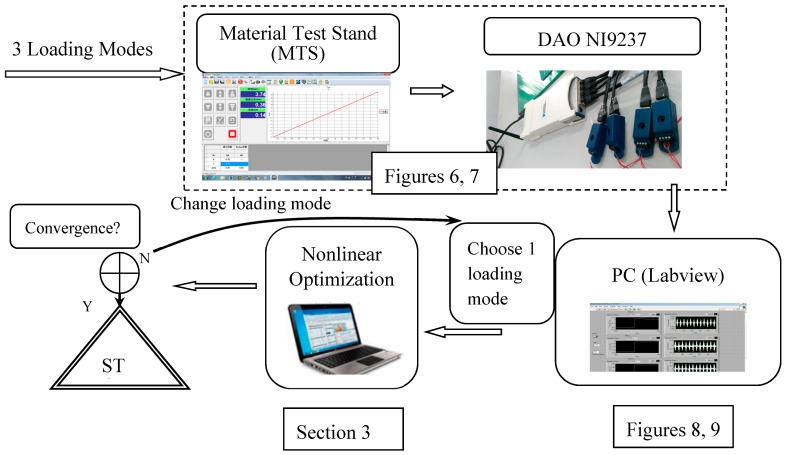
Hardware-in-the-loop simulation of hole/crack identification by using measured strains.

**Figure 6 materials-13-00424-f006:**
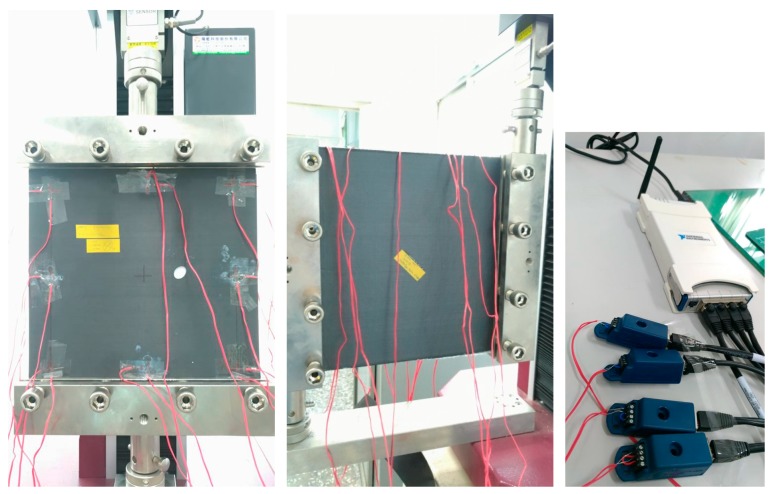
Material test stand (MTS) of hardware-in-the-loop simulation (HILS): Loading mode I and hole identification by using measured strains (**left**); loading mode II and crack (45°) identification (**middle**); Data Acquisition (DAQ) card NI9237—four channels strain measurement (**right**).

**Figure 7 materials-13-00424-f007:**
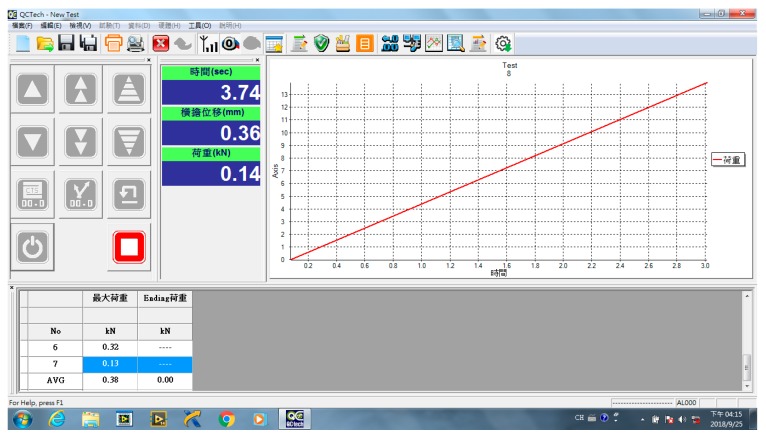
The relationship of loading and time in MTS.

**Figure 8 materials-13-00424-f008:**
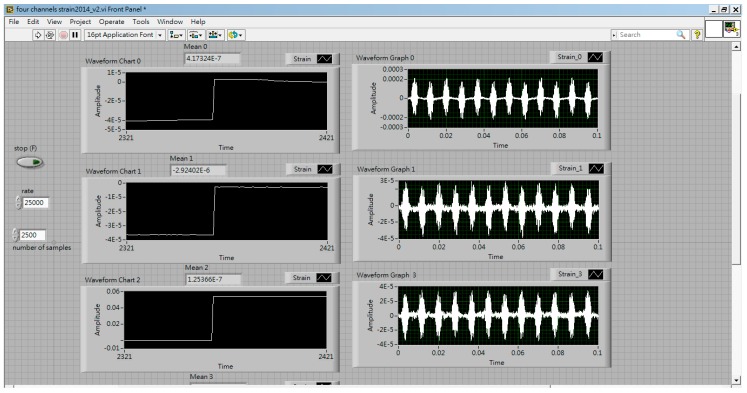
Front panel of strains measured from the composite plate.

**Figure 9 materials-13-00424-f009:**
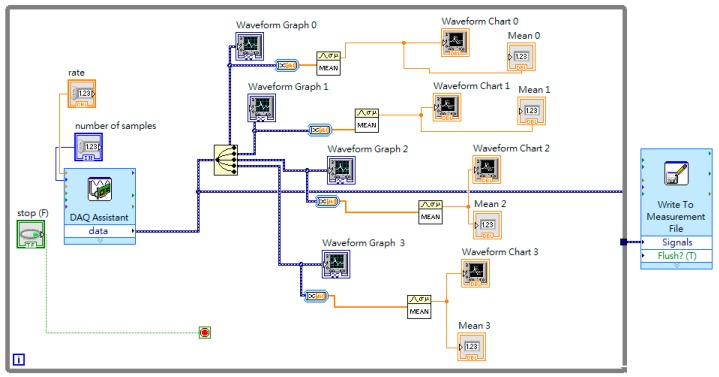
Block diagram of strains measured from the composite plate.

**Figure 10 materials-13-00424-f010:**
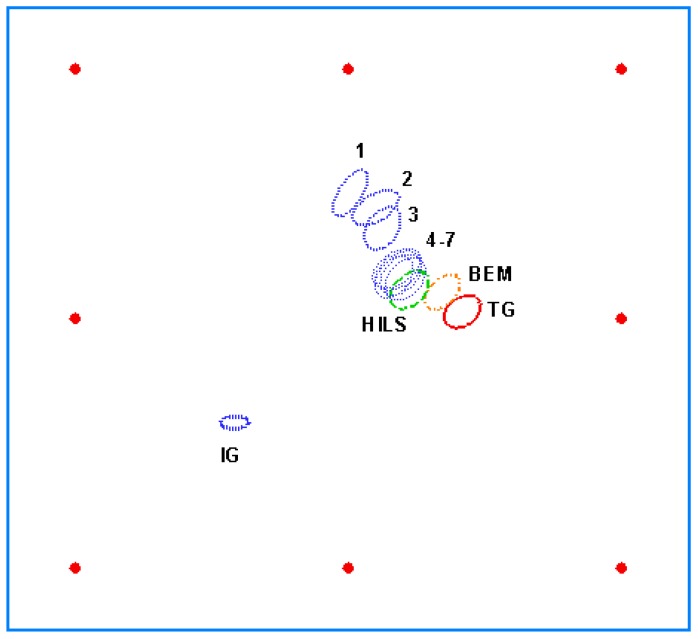
Procedures of hole identification by using multiple loading modes. Red points are sensor positions. The marked number signifies these seven identified processes.

**Figure 11 materials-13-00424-f011:**
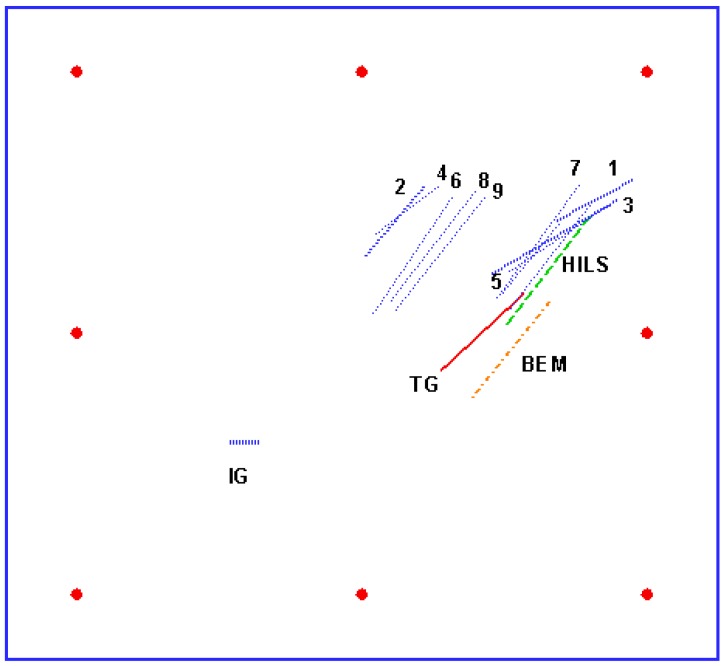
Procedures of crack identification by using multiple loading modes. Red points are sensor positions. No. 1–9 is the first nine identified results of different error tolerance *e*.

**Table 1 materials-13-00424-t001:** Calculated strains (unit μ) of the eight sensors in composite plate with a hole (BEM).

Sensor	Mode I			Mode II			Mode III	
	*ε_xx_*	*ε_xy_*	*ε_yy_*	*ε_xx_*	*ε_xy_*	*ε_yy_*	*ε_xz_*	*ε_yz_*
1	0.002755	−0.016474	0.010035	−0.008419	0.061529	0.107094	−0.000022	0.073782
2	−0.0007	−0.010732	0.011614	0.012631	0.04335	0.104055	0.00002	0.073868
3	−0.003461	−0.014707	0.013186	0.025224	−0.01437	0.074906	0.000092	0.073613
4	−0.000293	−0.019279	0.010883	0.031838	0.073769	0.037402	0.000256	0.073445
5	0.003659	−0.013187	0.011363	0.028607	0.078033	0.059318	−0.0001	0.073593
6	−0.001629	−0.012801	0.010576	0.013229	0.026622	0.044122	0.000191	0.074691
7	−0.003082	−0.0163	0.012225	−0.004738	0.034001	0.003485	0.000058	0.073386
8	−0.000087	−0.02075	0.010947	−0.013518	0.057657	0.042331	−0.000129	0.073479

**Table 2 materials-13-00424-t002:** Measured strains (unit μ) of the eight sensors in composite plate with a hole (NI 9237).

Sensor	Mode I		Mode II	
	*ε_xx_*	*ε_yy_*	*ε_xx_*	*ε_yy_*
1	0.002755	0.010035	−0.008419	0.107094
2	−0.0007	0.011614	0.012631	0.104055
3	−0.003461	0.013186	0.025224	0.074906
4	−0.000293	0.010883	0.031838	0.037402
5	0.003659	0.011363	0.028607	0.059318
6	−0.001629	0.010576	0.013229	0.044122
7	−0.003082	0.012225	−0.004738	0.003485
8	−0.000087	0.010947	−0.013518	0.042331
*ϕ_Μ_*	0.5148881		0.0112450	
*ϕ_Μ_* ^1^	345.44		130.07	

^1^ Hole geometry and location are *a* = 17.1 mm, *b =* 3 mm, *x =* 140 mm, *y =* 160 mm, and *θ* = 40°.

**Table 3 materials-13-00424-t003:** Hole identification of composite plate.

	Initial Guess	Target	BEM		HILS	
			Final Results	Error	Final Results	Error
*a* (mm)	6	9	9.82	9.11%	10.31	14.56%
*b* (mm)	3	6.5	5.83	10.31%	6.89	6.00%
*x* (mm)	100	200	191.31	4.35%	176.92	11.54%
*y* (mm)	100	153	162.41	6.14%	163.48	6.86%
*θ* (degree)	0	45	49.36	9.69%	50.34	11.87%

**Table 4 materials-13-00424-t004:** Calculated strains (unit μ) of the eight sensors in composite plate with a crack (BEM).

Sensor	Mode I			Mode II			Mode III	
	*ε_xx_*	*ε_xy_*	*ε_yy_*	*ε_xx_*	*ε_xy_*	*ε_yy_*	*ε_xz_*	*ε_yz_*
1	0.002846	−0.016149	0.010315	−0.00798	0.061701	0.106332	−0.000031	0.074738
2	−0.000644	−0.011064	0.011212	0.011702	0.042788	0.103302	0.000207	0.074555
3	−0.003739	−0.015733	0.012307	0.024874	−0.013262	0.074101	0.000161	0.073691
4	−0.00052	−0.018753	0.01125	0.031581	0.075858	0.036482	−0.000093	0.073047
5	0.00274	−0.017102	0.01039	0.028656	0.077735	0.060043	−0.0007	0.074093
6	−0.001008	−0.012255	0.011798	0.009771	0.022196	0.045539	0.001923	0.076213
7	−0.003314	−0.015822	0.012635	−0.00446	0.035829	0.003584	−0.000005	0.072115
8	−0.000247	−0.019797	0.011557	−0.012949	0.056025	0.042955	−0.000615	0.07379

**Table 5 materials-13-00424-t005:** Measured strains (unit μ) of the eight sensors in composite plate with a crack (NI 9237).

Sensor	Mode I		Mode II	
	*ε_xx_*	*ε_yy_*	*ε_xx_*	*ε_yy_*
1	0.002983	0.010901	−0.008167	0.106837
2	−0.000592	0.010472	0.011829	0.102974
3	−0.003813	0.011735	0.024934	0.073825
4	−0.000499	0.010825	0.032618	0.035281
5	0.002682	0.009899	0.029032	0.058753
6	−0.000943	0.011927	0.010067	0.044836
7	−0.003281	0.013023	−0.004732	0.003217
8	−0.000289	0.011973	−0.013028	0.043251
*ϕ_Μ_*	0.0602412		0.0189607	
*ϕ_Μ_* ^2^	171.75		105.01	

^2^ Crack geometry and location are *a* = 20 mm, *x =* 140 mm, *y =* 160 mm, and *θ* = 0°.

**Table 6 materials-13-00424-t006:** Crack identification of composite plate.

	Initial Guess	Target	BEM		HILS	
			Final Results	Error	Final Results	Error
*a* (mm)	6	25	28.67	14.68%	30.37	21.48%
*X* (mm)	80	200	211.66	5.85%	228.07	14.04%
*y* (mm)	80	150	141.33	5.80%	178.45	18.97%
*θ* (degree)	0	45	53.78	19.51%	55.03	22.29%

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
