# Peer review of "Hardware-In-The-Loop Simulations of Hole/Crack Identification in a Composite Plate"

_materials, 2020, doi:10.3390/ma13020424_

Round 1

Reviewer 1 Report

The paper presents the results of application of technology of hardware-in-the-loop simulations in damage detection in composite plate.

The objective of the study and the methodology followed are clearly outlined. The conclusions faithfully summarize the findings. My only concern is that the major contribution of this paper is unclear. Before I recommend the publication of the paper I would to ask the Authors to clearly indicate the novel aspects of their work.

Additional remarks:

-a lot of tables with data are not easy to follow by the reader - please, indicate the most important results (i.e. by highliting or using different colour)

-what is the influence of sensors configurations of the effectiness of the presented method?

-the tested specimen is characterized by a very small size. Please, describe the possible difficulties in the case of practical application of the method i.e. when the considered structure has larger size.

-what is the critical damage size which can be detected?

Author Response

First of all, the authors would like to thank the referees for their valuable and constructive comments.

Please see the attachment uploaded. 

Reviewer 2 Report

Questions and suggestions at the reading of the paper:

"A and B are eigenvectors of the material properties." Please, clarify the definition of these values. "For a straight crack, let b=0." Please, inform readers about assumption for the interaction of crack's edges.

Figs 1 and 2. Each considered loading mode produces different orientation of the plate's points displacement. These displacement can be registered by the sensors of different types. But description of Fig.1 does not contains any information about these sensors. Obviously, these sensors cannot be identical even for the tensile and shear loading, or two-dimensional rosette-like strain gauges should be used.

Formula (5). Description of the value "e" required.

235-238. The material of the plate under study was considered as quasi-isotropic or another structural anisotropy was assumed at the calculations?

Table 1. In above strings (184-187) the authors wrote "Therefore, only loading modes I and II is considered in this paper." But Table 1 contains two last rows were interlaminar shear strains are presented. How these calculated value are used at the crack's properties identification?

264-265. Please, clarify with more details the sentence "... crack identification is ... non-sensitive response of crack parameters to the strains around the plate".

Common recommendation:

It's not desirable to mention autopilot in Abstract because this gives the reader the first impression that the article claims to analyze the healthy state of the composite structure during flight. I recommend to exclude the word "autopilot" changing it by the another one. Some readers don't have access to the early published papers [15] and [26]. Therefore, it is desirable to describe with more details the part of the article between the strings 93 - 108. The type of the used sensors should be clarified. The choice of their number it would be useful to justify. To justify the ways of practical application of the method under consideration, it is important to give some examples of the real structures where the proposed technique can be used and indicate the possibility of applying test loads to create the pure shear and tension.

Conclusion:

This paper contains the very useful information that can be used at the design and maintenance of some load carrying composite structures. A minor revision is required.

Author Response

First of all, the authors would like to thank the referees for their valuable and constructive comments. Please see the uploaded attachment.

Round 2

Reviewer 1 Report

Dear Authors,

Thank you for your response, I recommend the publication of the article in the present form.